# Pl@ntNet-300K: a plant image dataset with high label ambiguity and a long-tailed distribution

**Camille Garcin**
IMAG, Univ Montpellier,
CNRS, Montpellier, France

**Alexis Joly**
Inria, LIRMM, Univ Montpellier,
CNRS, Montpellier, France

**Pierre Bonnet**
CIRAD, AMAP

**Jean-Christophe Lombardo**
Inria, LIRMM, Univ Montpellier,
CNRS, Montpellier, France

**Antoine Affouard**
Inria, LIRMM, Univ Montpellier,
CNRS, Montpellier, France

**Mathias Chouet**
Inria, LIRMM, Univ Montpellier,
CNRS, Montpellier, France

**Maximilien Servajean**
LIRMM, AMIS, UPVM,
Univ Montpellier, CNRS, Montpellier

**Titouan Lorieul**
Inria, LIRMM, Univ Montpellier,
CNRS, Montpellier, France

**Joseph Salmon**
IMAG, Univ Montpellier,
CNRS, Montpellier, France
Institut Universitaire de France (IUF)

## Abstract

This paper presents a novel image dataset with high intrinsic ambiguity and a long-tailed distribution built from the database of Pl@ntNet citizen observatory. It consists of 306,146 plant images covering 1,081 species. We highlight two particular features of the dataset, inherent to the way the images are acquired and to the intrinsic diversity of plants morphology: (i) the dataset has a strong *class imbalance*, *i.e.,* a few species account for most of the images, and, (ii) many species are *visually similar*, rendering identification difficult even for the expert eye. These two characteristics make the present dataset well suited for the evaluation of set-valued classification methods and algorithms. Therefore, we recommend two set-valued evaluation metrics associated with the dataset (*macro-average top-k accuracy* and *macro-average average-k accuracy*) and we provide baseline results established by training deep neural networks using the cross-entropy loss.

## 1 Introduction

When classifying images, we are faced with two main types of uncertainties [Der Kiureghian and Ditlevsen, 2009]: (i) the *aleatoric* uncertainty that arises from the intrinsic randomness of the underlying process, which is considered irreducible, and (ii) the *epistemic* uncertainty that is caused by a lack of knowledge and is considered to be reducible with additional training data. In modern real-world applications, these two types of uncertainties are particularly difficult to handle. The ever-growing number of classes to distinguish tends to increase the class overlap (and thus the aleatoric uncertainty), and, on the other hand, the long-tailed distribution makes it difficult to learn the less populated classes (and thus increase the epistemic uncertainty). The presence of these two uncertainties is a central motivation for the use of set-valued classifiers, *i.e.,* classifiers returning a set of

35th Conference on Neural Information Processing Systems (NeurIPS 2021) Track on Datasets and Benchmarks.

candidate classes for each image [Chzhen et al., 2021]. Although there are several datasets in the literature that have visually similar classes [Nilsback and Zisserman, 2008, Maji et al., 2013, Yang et al., 2015, Russakovsky et al., 2015], most of them do not aim to retain both the epistemic and the aleatoric ambiguity present in real-world data.

In this paper, we propose a dataset designed to remain representative of real-life ambiguity, making it well suited for the evaluation of set-valued classification methods. This dataset is extracted from real-world images collected as part of the Pl@ntNet project [Affouard et al., 2017], a large-scale citizen observatory dedicated to the collection of plant occurrences data through image-based plant identification. The key feature of Pl@ntNet is a mobile application that allows citizens to send a picture of a plant they encounter and get a list of the most likely species for that photo in return. The application is used by more than 10 million users in about 170 countries and is one of the main data publishers of GBIF[1], an international platform funded by the governments of many countries around the world to provide free and open access to biodiversity data. Another essential feature of Pl@ntNet is that the training set used to train the classifier is collaboratively enriched and revised. Nowadays, Pl@ntNet covers over 35K species illustrated by nearly 12 million validated images.

The entire Pl@ntNet database would be an ideal candidate for the evaluation of set-valued classification methods. However, it is far too large to allow for widespread use by the machine learning community. Extracting a subsample from it must be done with care as we want to preserve the uncertainty naturally present in the whole database. The dataset presented in this paper is constructed by retaining only a subset of the genera of the entire Pl@ntNet database (sampled uniformly at random). All species belonging to the selected genera are then retained. Doing so maintains the original ambiguity as species in the same genus are likely to be visually similar and to share common visual features.

The rest of the paper is organized as follows. We first introduce the set-valued classification framework in Section 2, focusing on two special cases: top-$k$ classification and average-$k$ classification. In Section 3, we describe the construction procedure of the Pl@ntNet-300K dataset and show that it contains a large amount of ambiguity. Next, we present in Section 4 the metrics of interest for the dataset and propose benchmark results for these metrics, obtained by training several neural networks architectures. In Section 5, we compare Pl@ntNet-300K to several existing datasets. In Section 6, we discuss possible uses of Pl@ntNet-300K. Finally, we provide the link to the dataset in Section 7 before concluding.

## 2    Set-valued classification

We adopt the classical statistical setup of multi-class classification. Let $L$ be the number of classes. We denote by $[L]$ the set $\{1, \ldots, L\}$ and by $\mathcal{X}$ the input space. Random couples of images and labels $(X, Y) \in \mathcal{X} \times [L]$ are assumed to be generated *i.i.d.* by an unknown joint distribution $\mathbb{P}$. Note that only one label is associated with each image, which differs from the multi-label setting [Zhang and Zhou, 2014]: P@ntNet-300K is composed of images containing a single specimen of a plant, so there is only one true label per image. For some plant images, predicting the correct class label (the correct species) does not present much difficulty (consider for instance a common species very distinctive from other species). For other images, however, classifying the photographed specimen with a high degree confidence is a much harder task, because some species differ only in subtle visual features (see Figure 4). In these cases, it is desirable to provide the user with a list of likely species corresponding to the image. We thus need a classifier able to produce sets of classes, also known as a set-valued classifier in the literature [Chzhen et al., 2021]. A set-valued classifier $\Gamma$ is a function mapping the feature space $\mathcal{X}$ to the set of all subsets of $[L]$ (which we denote by $2^{[L]}$). Using these notations, we thus have $\Gamma : \mathcal{X} \rightarrow 2^{[L]}$ instead of $\Gamma : \mathcal{X} \rightarrow [L]$ for the classical setting in which the predictor can only predict a single class. Our goal is to build a classifier with low risk, defined as $\mathbb{P}(Y \notin \Gamma(X))$. However, it is not desirable to simply minimize the risk: a set-valued classifier that always returns all classes achieves zero risk but is useless. On the other hand, a classifier is most useful if it returns only the most likely classes given a query image. Therefore, a quantity of interest will be $|\Gamma(x)|$, the number of classes returned by the classifier $\Gamma$, given an image $x \in \mathcal{X}$.

---

[1]https://www.gbif.org/

In this section we will examine two optimization problems that lead to different set-valued classifiers. Both of them aim to minimize the risk, but they differ in the way they constrain the set cardinality: either pointwise or on average.

For $x \in \mathcal{X}$ and $l \in [L]$, we define the conditional probability $p_l(x) := \mathbb{P}(Y = l \mid X = x)$, and estimators of these quantities will be denoted by $\hat{p}_l(x)$. In the following, $k \in [L]$. Finally, for $x \in \mathcal{X}$, we define $\text{top}_p(x, k)$ as the set containing the $k$ indexes corresponding to the $k$ largest values of $\{p_l(x)\}_{l \in [L]}$.

The simplest constraint is to require that the number of returned classes is less than $k$ for each input. This results in the following top-$k$ error [Lapin et al., 2015] minimization problem:

$$\Gamma^*_{\text{top-k}} \in \arg\min_{\Gamma} \mathbb{P}(Y \notin \Gamma(X))$$
$$\text{s.t. } |\Gamma(x)| \leq k, \ \forall x \in \mathcal{X} \ . \tag{1}$$

The closed form solution to (1) exists and is equal to [Lapin et al., 2017]:

$$\Gamma^*_{\text{top-k}}(x) = \text{top}_p(x, k) \ . \tag{2}$$

Yet, this is not practical since we do not know the distribution $\mathbb{P}$ and thus $p_l(x)$. However, if we have an estimator $\hat{p}_l(x)$ of $p_l(x)$, we can naturally derive the plug-in estimator: $\widehat{\Gamma}_{\text{top-k}} = \text{top}_{\hat{p}}(x, k)$. While the *top-k accuracy* is often reported in benchmarks, only a few works aim to directly optimize this metric [Lapin et al., 2015, 2016, 2017, Berrada et al., 2018]. An obvious limitation of top-$k$ classification is that $k$ classes are returned for every data sample, regardless of the difficulty of classifying that sample. Average-$k$ classification allows for more adaptivity. In that setting, the constraint on the size of the predicted set is less restrictive and must be satisfied only on average, leading to:

$$\Gamma^*_{\text{average-k}} \in \arg\min_{\Gamma} \mathbb{P}(Y \notin \Gamma(X))$$
$$\text{s.t. } \mathbb{E}_X |\Gamma(X)| \leq k \ . \tag{3}$$

The closed form solution is derived in [Denis and Hebiri, 2017]:

$$\Gamma^*_{\text{average-k}}(x) = \{l \in [L] : p_l(x) \geq G^{-1}(k)\} \ , \tag{4}$$

where the function $G$ is defined as: $\forall t \in [0, 1], \quad G(t) = \sum_{l=1}^{L} \mathbb{P}(p_l(X) \geq t)$, and $G^{-1}$ refers to the generalized inverse of $G$, namely $G^{-1}(u) = \inf\{t \in [0, 1] : G(t) \leq u\}$.

Note that if we define the classifier $\Gamma_t$ by: $\forall x \in \mathcal{X}, \Gamma_t(x) = \{l \in [L], p_l(x) \geq t\}$, then $G(t)$ is the average number of classes returned by $\Gamma_t$: $G(t) = \mathbb{E}_X |\Gamma_t(X)|$. From (4) we see that the optimal classifier corresponds to a thresholding operation: all classes having a conditional probability greater than $G^{-1}(k)$ are returned, with the threshold chosen so that $k$ classes are returned on average. To compute a plug-in counterpart, we just need to estimate the threshold on a calibration set such that on average on that set, $k$ classes are returned. For technical details, we refer the reader to Denis and Hebiri [2017].

## 3 Dataset

### 3.1 Label validation and data cleaning

Label validation is based on a weighted majority voting algorithm taking as input the labels proposed by Pl@ntNet users with an adaptive weighting principle according to the user's expertise and commitment. Thus, a single trusted annotator can be enough to validate an image label. On the other hand, images whose labels are proposed by several novice users may not be validated because they do not have sufficient weight. The technical details of this algorithm can be found in the supplementary material. At the time of the construction of Pl@ntNet-300K, the total number of annotators in the Pl@ntNet database was equal to 2,079,003. The average number of annotators per image is equal to 2.03.

In addition to the label validation procedure, Pl@ntNet's pipeline includes other data cleaning procedures: (i) automated filtering of inappropriate or irrelevant content (faces, humans, animals, buildings, etc.) using a CNN and user reports, and (ii) filtering on image quality (evaluated by users).

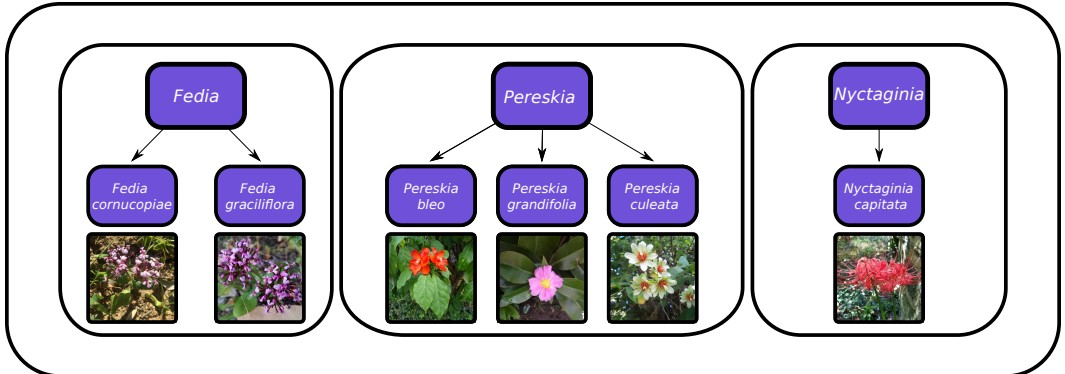

Figure 1: Genus taxonomy: we display three genera present in the proposed dataset—*Fedia*, *Pereskia* and *Nyctaginia*—which contain respectively two, three and one species.

## 3.2 Construction of Pl@ntNet-300K

In taxonomy, species are organized into genera, with each genus containing one or more species, and the different genera do not overlap, as illustrated in Figure 1.

Instead of retaining randomly selected species or images from the entire Pl@ntNet dataset, we choose to retain randomly selected genera and keep all species belonging to these genera. This choice aims to preserve the large amount of ambiguity present in the original database, as species belonging to the same genus tend to share visual features. The dataset presented in this paper is constructed by sampling uniformly at random only 10% of the genera of the whole Pl@ntNet database.

We then retain only species with more than 4 images, resulting in a total of 303 genera and $L = 1{,}081$ species. The images are divided into a training set, a validation set and a test set.[2] For each species, 80% of the images are placed in the training set ($n_{train} = 243{,}916$), 10% in the validation set ($n_{val} = 31{,}118$), and 10% in the test set ($n_{test} = 31{,}112$), with at least one image of each species in each set. More formally, given a class $j$ containing $n_j$ images, $n_{val,j} = \lceil 0.1 \times n_j \rceil$, $n_{test,j} = \lceil 0.1 \times n_j \rceil$ and $n_{train,j} = n_j - n_{val,j} - n_{test,j}$. This represents a total of $n_{tot} = n_{train} + n_{val} + n_{test} = 306{,}146$ color images. The average image size is (570, 570, 3), ranging from (180, 180, 3) to (900, 900, 3). The construction of the dataset preserves the class imbalance. To show this, we plot the Lorenz curves [Gastwirth, 1971] of the entire Pl@ntNet dataset and that of the Pl@ntNet-300K dataset in Figure 2.

## 3.3 Epistemic (model) uncertainty

Epistemic uncertainty refers mainly to the lack of data necessary to properly estimate the conditional probabilities. In Pl@ntNet, the most common species are easily observed by users in the wild and therefore represent a large fraction of the images, while the rarest species are harder to find and therefore less frequent in the database. In Figure 2, we see that 80% of the species (the ones with the lowest number of images) account for only 11% of the total number of images. Hence, training machine learning models is challenging for such a dataset, since for many classes the model only has a handful of images to train on, making identification difficult for these species.

In addition to the long-tailed distribution issue, epistemic uncertainty also arises from the high intra-species variability. Plants may take on different appearances depending on the season (*e.g.,* , flowering time). Furthermore, a user of the application may photograph only a part of the plant (for instance, the trunk and not the leaves). As a last example, flowers belonging to the same species can have different colors. Figure 3 shows some examples of these phenomena which contribute to high intra-class variability, making it more challenging to model the species.

---

[2]The division is performed at the species level due to the long-tailed distribution.

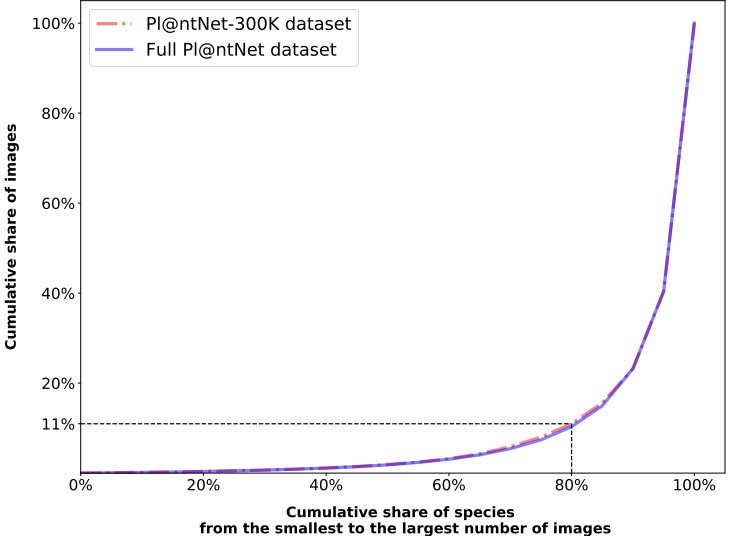

Figure 2: Lorenz curves of the Pl@ntNet database and the proposed dataset. Note that, for fair comparison, we discard species with less than 4 images in the Pl@ntNet database.

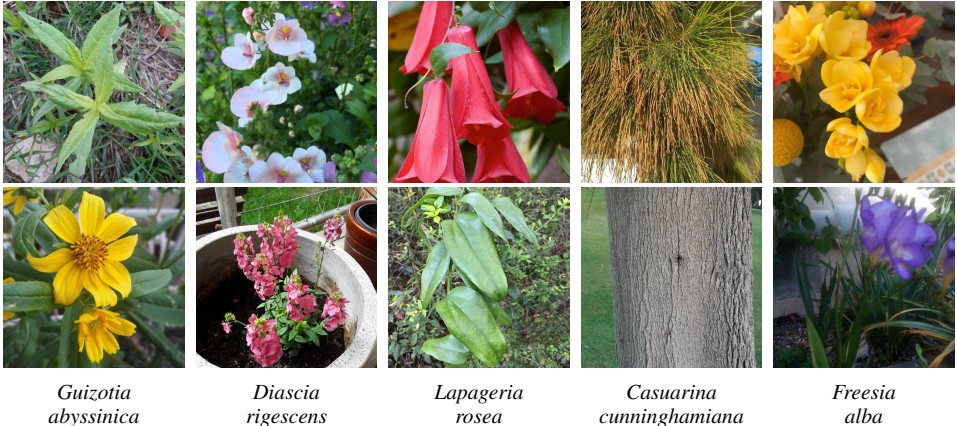

| *Guizotia abyssinica* | *Diascia rigescens* | *Lapageria rosea* | *Casuarina cunninghamiana* | *Freesia alba* |

Figure 3: Examples of visually different images belonging to the same class.

## 3.4 Aleatoric (data) uncertainty

In our case, the source of aleatoric uncertainty mostly resides in the limited information we are given to make a decision (assign a label to a plant). Some species, especially those belonging to the same genus, can be visually very similar. For example, consider the case where two species produce the same flowers but different leaves, typically because they have evolved differently from the same parent species. If a person photographs only the flower of a specimen of one of the two species, then it will be impossible, even for an expert, to know which species the flower belongs to. The discriminative information is not present in the image.

The combination of this irreducible ambiguity with images of non-optimal quality (non-adapted close-up, low-light conditions, etc.) results in pairs of images that belong to different species but are difficult or even impossible to distinguish, see Figure 4 for illustration. In this figure, we show the ambiguity between pairs of species, but we could find similar examples involving a larger number of species. Thus, even an expert botanist might fail to assign a label to such pictures with certainty. This is embodied by $p_l(x)$ : given an image, multiple classes are possible.

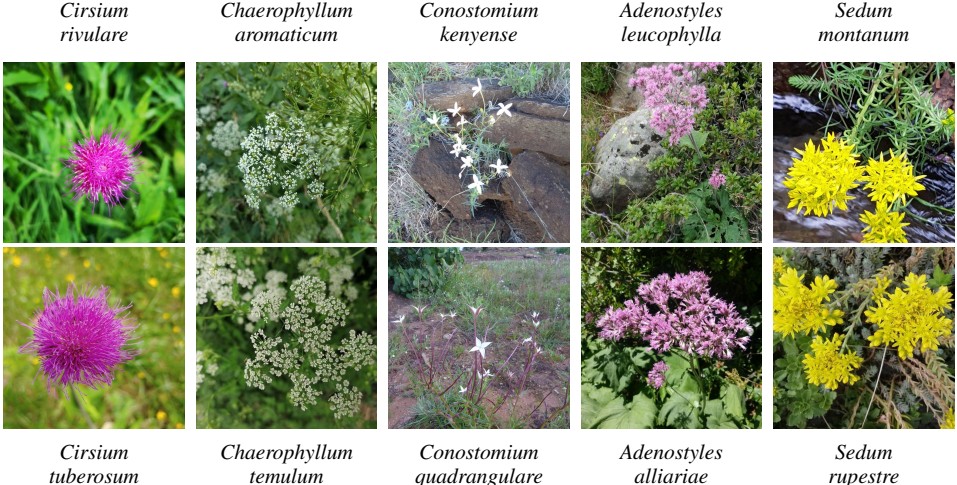

| *Cirsium rivulare* | *Chaerophyllum aromaticum* | *Conostomium kenyense* | *Adenostyles leucophylla* | *Sedum montanum* |
| *Cirsium tuberosum* | *Chaerophyllum temulum* | *Conostomium quadrangulare* | *Adenostyles alliariae* | *Sedum rupestre* |

Figure 4: Examples of visually similar images belonging to two different classes.

# 4 Evaluation

## 4.1 Metric

We consider two main metrics to evaluate set valued predictors on Pl@ntNet-300K: *top-k accuracy* and *average-k accuracy*. Let $S$ denote a set of $n$ (input, label) pairs: $S = \{(x_1, y_1), (x_2, y_2), \ldots, (x_n, y_n)\}$. *Top-k accuracy* [Lapin et al., 2016] is a widely used metric often reported in benchmarks. *Average-k accuracy* is a much less common metric that derives from average-$k$ classification [Denis and Hebiri, 2017]:

$$average\text{-}k \; accuracy(S) = \frac{1}{n} \sum_{(x_i, y_i) \in S} \mathbb{1}_{[y_i \in \widehat{\Gamma}_{\text{average-k}}(x_i)]} \; \text{s.t.} \; \frac{1}{n} \sum_{(x_i, y_i) \in S} |\widehat{\Gamma}_{\text{average-k}}(x_i)| \leq k \; , \quad (5)$$

where $\widehat{\Gamma}_{\text{average-k}}$ is a set-valued classifier constructed using the training data.

For Pl@ntNet-300K, both *top-k accuracy* and *average-k accuracy* mainly reflect the performance of the set-valued classifier on the few classes which represent most of the images. If we wish to capture the ability of a set-valued classifier to return pertinent set of species for all classes, we will examine *macro-average top-k accuracy* and *macro-average average-k accuracy* which simply consist in computing respectively *top-k accuracy* and *average-k accuracy* for each class, and then computing the average over classes. For *macro-average average-k accuracy*, the constraint on the average size of the set must hold for the entire set $S$.

To derive both classifiers, one can first obtain an estimate of the conditional probabilities $\hat{p}_l(x)$ and then derive the plug-in classifiers, as explained in Section 2. Our hope is for the Pl@ntNet-300K dataset to encourage novel ways to derive the set-valued classifiers $\widehat{\Gamma}_{\text{top-k}}$ and $\widehat{\Gamma}_{\text{average-k}}$ to optimize respectively the *top-k accuracy* and the *average-k accuracy*. Notice that a few works already propose methods to optimize the *top-k accuracy* [Lapin et al., 2015, 2016, 2017, Berrada et al., 2018].

## 4.2 Baseline

This section provides a baseline evaluation of the plug-in classifiers. We train several deep neural networks with the cross-entropy loss: ResNets [He et al., 2016], DenseNets, [Huang et al., 2017], InceptionResNet-v2 [Szegedy et al., 2017], MobileNetV2 [Sandler et al., 2018], MobileNetV3 [Howard et al., 2019], EfficientNets [Tan and Le, 2019], Wide ResNets [Zagoruyko and Komodakis, 2016], AlexNet [Krizhevsky et al., 2012], Inception-v3 [Szegedy et al., 2016], Inception-v4 [Szegedy et al., 2017], ShuffleNet [Zhang et al., 2018], SqueezeNet [Iandola et al., 2016], VGG [Simonyan and Zisserman, 2015] and Vision Transformer [Dosovitskiy et al., 2021].

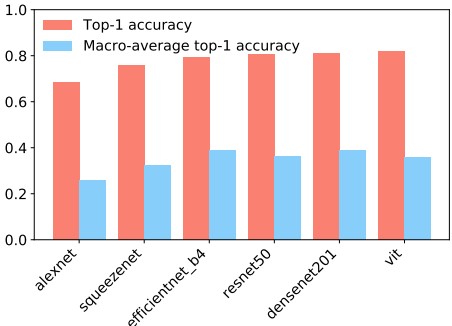

Figure 5: Pl@ntNet-300K test *top*-1 *accuracy* and *macro-average top*-1 *accuracy* for several neural networks.

| Number of images | Mean bin accuracy |
|---|---|
| $0 - 10$ | $0.09$ |
| $10 - 50$ | $0.35$ |
| $50 - 500$ | $0.59$ |
| $500 - 2000$ | $0.79$ |
| $> 2000$ | $0.93$ |

Table 1: Test accuracy depending on the number of images per class in the training set. Obtained with a ResNet50.

All models are pre-trained on ImageNet. During training, images are resized to 256 and a random crop of size $224 \times 224$ is extracted. During test time, we take the centered crop.

The models are optimized with SGD with a momentum of 0.9 with the Nesterov acceleration [Ruder, 2016]. We use a batch size of 32 for all models and a weight decay of $1.10^{-4}$. The number of epochs, initial learning rate and learning rate schedule used for each model can be found in the supplementary material. For the plug-in classifier $\widehat{\Gamma}_{\text{average-k, plug-in}}$, we compute the threshold $\lambda_{val}$ on the validation set and use that same threshold to compute the *average-k accuracy* on the test set.

### 4.3 Difficulty of Pl@ntNet-300K

Figure 5 highlights the significant gap between Pl@ntNet-300K *top*-1 *accuracy* and *macro-average top*-1 *accuracy*. This is a consequence of the long-tailed distribution: the few classes that represent most of the images are easily identified, which results in high *top*-1 *accuracy*. However, this seemingly high *top*-1 *accuracy* is misleading, as models struggle with classes with few images (which are a majority, see Figure 2). This effect is illustrated in Table 1, which shows that the *top*-1 *accuracy* depends strongly on the number of images in the class.

Figure 8a shows the correlation between Pl@ntNet-300K *macro-average top*-1 *accuracy* and ImageNet *macro-average top*-1 *accuracy* (note that as the ImageNet test set is balanced, *top*-1 *accuracy* and *macro-average top*-1 *accuracy* coincide). As expected, the two metrics are positively correlated: deep networks allowing to model complex features work well both on ImageNet and Pl@ntNet-300K. Interestingly, due the difference between the two datasets (long-tailed distribution, class ambiguity, . . . ), some models which perform similarly on ImageNet yield very different on Pl@ntNet-300K (inception_v3, densenet201), and vice versa.

In Figure 8a we can notice that ImageNet *macro-average top*-1 *accuracy* and Pl@ntNet-300K *macro-average top*-1 *accuracy* vary at different scales: the former goes up to 80% while the latter does not exceed 40%, making Pl@ntNet-300K a challenging dataset with both epistemic and aleatoric uncertainty at play.

This can can also be seen in Figure 6: some models reach a *macro-average average*-5 *accuracy* of 97% for ImageNet, while that metric does not exceed 80% for Pl@ntNet-300K, which suggests that progress could be made with appropriate learning strategies.

To support that claim, we asked a botanist to label a mini dataset extracted from the Pl@ntNet-300K test set. The dataset is constructed as follows: we extract all species from two groups (*Crotalaria* and *Lupinus*), and select at most 5 images per species (randomly sampled). This results in 83 images. The botanist was asked to provide a set of possible species for each image. This results in an error rate of 20.5% for an average of 4,1 species returned (ranging from 1 to 10). We compare the botanist performance with that of several neural networks by calibrating the conditional probabilities' threshold to obtain on average 4,1 species on the mini-dataset. The results are reported in Figure 7, and show that the gap between the botanist error rate and the best performing model is

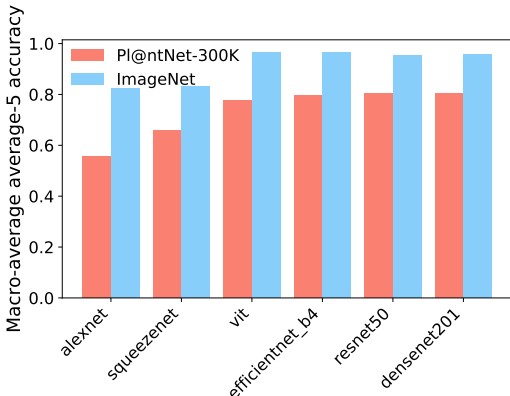

Figure 6: Pl@ntNet-300K vs ImageNet *macro-average average*-5 *accuracy* for several models (evaluated on the test set).

Figure 7: Error rate on the mini test set of an expert compared to several neural networks. All models and the expert return on average 4.1 species on the mini test set.

large (from 0.2 to 0.33), which suggests that there is room for improvement in the performance of average-$k$ classifiers.

### 4.4 Top-$k$ vs Average-$k$

From Equation (1) and (3), it is clear that the Bayes average-$k$ classifier has a lower risk than the Bayes top-$k$ classifier. Therefore, a model that accurately estimates the conditional probabilities should yield a better *average-k accuracy* than *top-k accuracy*. This is what can be observed in Figure 8b which shows the correlation between *macro-average top*-5 *accuracy* and *macro-average average*-5 *accuracy*. As expected, the two metrics are positively correlated. However, the relationship does not appear to be trivial and Figure 8b shows models with similar *macro-average average*-5 *accuracies* having very different *macro-average top*-5 *accuracy* and vice versa. For an in-depth comparison of average-$k$ classification and top-$k$ classification, we refer the reader to Lorieul [2020]. Both metrics are of their own interest and deserve a specific treatment as they capture two different settings: *top-k accuracy* evaluates the performance of a classifier which systematically returns $k$ classes, while *average-k accuracy* evaluates the performance of a classifier which returns sets of varying size (depending on the input), with the constraint to return $k$ classes on average.

### 4.5 Evaluation of existing set-valued classification methods

To the best of our knowledge, there is no existing loss designed to specifically optimize *average-k accuracy*. For top-$k$ classification, the most recent loss designed to optimize *top-k accuracy* is the one by Berrada et al. [2018]. We report the *top-5 accuracy* obtained by training this loss with $k = 5$ on Pl@ntNet-300K in the supplementary material. The results are close with what is obtained with the cross entropy loss. However, this topic is still open research and our hope in releasing Pl@ntNet-300K is precisely to encourage novel methods for optimizing such metrics.

## 5 Related work

Fined-Grained Visual Categorization (FGVC) is about discriminating visually similar classes. In order to better learn fine-grained classes, several approaches have been proposed by the FGVC community, including multi-stage metric learning [Qian et al., 2015], high order feature interaction [Lin et al., 2015, Cui et al., 2017], and different network architectures [Fu et al., 2017, Ge et al., 2016]. However, these approaches focus on optimizing *top-1 accuracy*. Set-valued classification, on the other hand, consists in returning more than a single class to reduce the error rate, with a constraint on the number of classes returned. Therefore, FGVC and set-valued classification methods are not mutually exclusive but rather complementary.

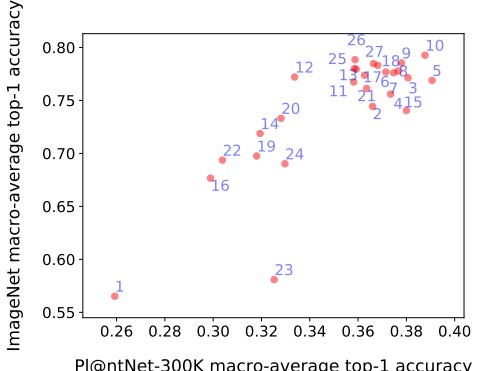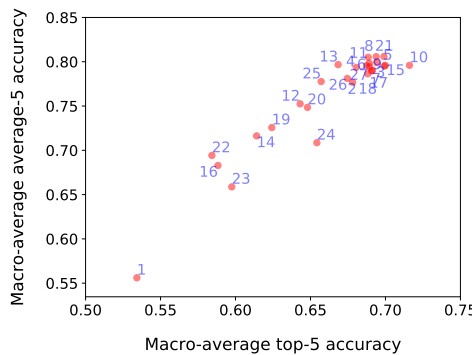

(a) ImageNet *macro-average top*-1 *accuracy* vs. Pl@ntNet-300K *macro-average top*-1 *accuracy* (evaluated on the test set).

(b) Pl@ntNet-300K *macro-average average*-5 *accuracy* vs. *macro-average top*-5 *accuracy* (evaluated on the test set).

Figure 8: Benchmark for several popular deep neural network architectures[3].

Several FGVC datasets, which exhibit visually similar classes, have been made publicly available by the community. They cover a variety of domains: aircraft [Maji et al., 2013], cars (Compcars [Yang et al., 2015], Census cars [Gebru et al., 2017]), birds (CUB200 [Welinder et al., 2010]), flowers (Oxford flower dataset [Nilsback and Zisserman, 2008]). However, most of these datasets focus exclusively on proposing visually similar classes (aleatoric uncertainty) with a limited amount of epistemic uncertainty. This is the case for balanced datasets which have approximately the same number of images per class, or with small intra-class variability such as aircraft and cars datasets, where most examples within a class are nearly the same except for angle, lightning, etc. ImageNet [Russakovsky et al., 2015] has several visually similar classes, organized in groups : it contains many bird species and dog breeds. However, these groups of classes are very different: dogs, vehicles, electronic devices, etc. Besides, ImageNet does not exhibit a strong class imbalance. Several of these datasets were constructed by web-scraping, which can be prone to noisy labels and low quality images. Most similar to our dataset is the iNat2017 dataset [Horn et al., 2018]. It contains images from the citizen science website iNaturalist. The images, posted by naturalists, are validated by multiple citizen scientists. The iNat2017 dataset contains over 5000 classes that are highly unbalanced. However, iNat2017 does not only focus on plants but proposes several other 'super-classes' such as *Fungi*, *Reptilia*, *Insecta*, etc. Moreover, the authors selected all classes with a number of observations greater than 20, whereas we choose to randomly sample 10% of the genera of the entire Pl@ntNet database and keep all species belonging to these groups with a number of observations greater than 4. We argue that keeping all species of the same genus maximizes aleatoric uncertainty, because species belonging to the same genus tend to share visual features. Finally, a plant disease dataset is introduced in [Sladojevic et al., 2016], containing 4483 images downloaded from the web spread across 15 classes. This is a very different scale than Pl@ntNet-300K. We summarize the properties of the mentioned datasets in Table 2.

# 6 Possible uses of Pl@ntNet-300K

Although we are convinced of the need to design new set-valued methods due to the ever increasing amount of classes to discriminate, the properties of Pl@ntNet-300K described in Section 3 make it an ideal candidate for various other tasks. The strong class imbalance can be used by researchers to evaluate new algorithms specifically designed for tackling class imbalance [Zhou et al., 2020, Cao et al., 2019]. Pl@ntNet-300K contains a large amount of aleatoric uncertainty resulting from many visually similar classes. It can therefore be used as a FGVC dataset to evaluate methods that aim

---

[3]The architectures chosen are: alexnet (1), densenet121 (2), densenet161 (3), densenet169 (4), densenet201 (5), efficientnet_b1 (6), efficientnet_b1 (7), efficientnet_b2 (8), efficientnet_b3 (9), efficientnet_b4 (10), inception_resnet_v2 (11), inception_v3 (12), inception_v4 (13), mobilenet_v2 (14), mobilenet_v3_large (15), mobilenet_v3_small (16), resnet101 (17), resnet152 (18), resnet18 (19), resnet34 (20), resnet50 (21), shufflenet (22), squeezenet (23), vgg11 (24), vit_base_patch16_224 (25), wide_resnet101_2 (26), wide_resnet50_2 (27).

Table 2: Comparison of several datasets with Pl@ntNet-300K. "Focused domain" indicates whether the dataset is made up of a single category (*i.e.,* cars) and "Ambiguity preserving sampling" indicates whether in the construction of the dataset, all classes belonging to the same parent in the class hierarchy were kept or not (in our case, the parent corresponds to the genus level).

| | Human-in-the-loop labeling | Long-tailed distribution | Intra-class variability | Focused domain | Ambiguity preserving sampling |
|---|---|---|---|---|---|
| Plant disease dataset | ✗ | ✗ | ✗ | ✓ | ✗ |
| CUB200 | ✗ | ✗ | ✗ | ✓ | ✗ |
| Oxford flower dataset | ✗ | ✗ | ✓ | ✓ | ✗ |
| Aircraft dataset | ✓ | ✗ | ✗ | ✓ | ✗ |
| Compcars | ✗ | ✗ | ✗ | ✓ | ✓ |
| Census cars | ✗ | ✗ | ✗ | ✓ | ✓ |
| ImageNet | ✗ | ✗ | ✓ | ✗ | ✗ |
| iNat2017 | ✓ | ✓ | ✓ | ✗ | ✗ |
| Pl@ntNet-300K | ✓ | ✓ | ✓ | ✓ | ✓ |

to optimize *top*-1 *accuracy* for such datasets, see for instance [Lin et al., 2015, Fu et al., 2017, Cui et al., 2017]. Finally, in this paper we do not use the genus information and thus do not exploit the hierarchical structure of the problem. In this sense, we adopt the flat classification approach described in [Silla and Freitas, 2011]. This is consistent with ImageNet [Russakovsky et al., 2015] or CIFAR-100 [Krizhevsky, 2009], where a hierarchy does exist but is rarely used in benchmarks. However, researchers are free to use the genus information to evaluate hierarchical classification methods on Pl@ntNet-300K.

# 7 Data access and additional resources

The Pl@ntNet-300K dataset can be found here:

https://doi.org/10.5281/zenodo.5645731.

It is organized in three folders named "`train`", "`val`" and "`test`". Each of these folders contains $L = 1,081$ subfolders. We provide the correspondence between the names of the subfolders and the names of the classes in the file "`plantnet300K_species_id_2_name.json`". We also provide a metadata file named "`plantnet300K_metadata.json`" containing for each image the following information: the species identifier (class), the organ of the plant (flower, leaf, bark, . . . ), the author's name, the license and the split (*i.e.,* train, validation or test set). A github repository containing the code to reproduce the experiments of this paper (where potential issues related to the dataset can be reported too) is available at: https://github.com/plantnet/PlantNet-300K/.

# 8 Conclusion

In this paper, we share and discuss a novel plant image dataset, called Pl@ntNet-300K, obtained as a subset of the entire Pl@ntNet database and intended primarily for evaluating set-valued classification methods. Unlike previous datasets, Pl@ntNet-300K is designed to preserve the high level of ambiguity across classes of the initial real-world dataset as well as its long-tailed distribution. To evaluate set-valued predictors on Pl@ntNet-300K, we investigate two different metrics: *macro-average top-$k$ accuracy* and *macro-average average-$k$ accuracy*, which is a more challenging task requiring to predict sets of various size but still equal to $k$ on average. Our results suggest that there is room for new set-valued prediction methods that would improve the performance of average-$k$ classifiers. We hope that Pl@ntNet-300K can serve as a reference dataset for this problem, which is our main motivation for releasing and sharing it with the community. We also stress that Pl@ntNet-300K can also be used to evaluate new methods for long-tailed classification and FGVC.

## Acknowledgments

This work was partially funded by the ANR CaMeLOt ANR-20-CHIA-0001-01. It has received funding from the European Union's Horizon 2020 research and innovation program under grant agreement No 863463 (Cos4Cloud project).

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
