# OpenReview forum: "Pl@ntNet-300K: a plant image dataset with high label ambiguity and a long-tailed distribution"
_NeurIPS.cc/2021/Track/Datasets_and_Benchmarks/Round2 — NeurIPS 2021 Datasets and Benchmarks Track (Round 2)_

### Official Review · Reviewer_iTKL · 2021-09-05
**Dataset for set-classifiers**

**Rating:** 5
**Confidence:** 4
**Correctness:** Yes.

**Strengths:**

The baseline models for evaluation are comprehensive and provide insight into the different degrees of success for each model. The comparison to an expert botanist shows room for improvement for existing models.

Evaluation metric and analysis is well-defined with a justification for why the metrics were chosen.

Comparison to prior work shows that the paper's motivation is well-founded and is distinguishable from past datasets. An appropriate dataset split and acknowledgement of the task's difficulty was qualitatively shown.


**Weaknesses:**

I am mainly concerned about the significance of contribution, as the main contribution of the dataset is to randomly sample 10% of genera from the Pl@ntNet database. The evaluation metrics recommended are canonical classification metrics.

In addition, there doesn't seem to be data cleaning involved. Though the authors retain uncertainty in the dataset, the variability involved in users taking different angled pictures of plants, different zoom/crop of plants, plants taking on different appearances depending on seasons (flowering time), is too high. Additional subsets of the dataset where this is cleaned, separated into different versions and accounted for (for example, Pl@ntNet-cropped, etc), is recommended.

The authors also acknowledge that if a person photographs only the flower of a specimen of one of the two similar species, then it will be impossible, even for an expert, to know which species the flower belongs to. Though this is a reasonable task for set-classifiers, it seems misleading then for the authors to report mean top-1 accuracy for an impossible task in Section 4.3. When even the error rate for a botanist is above 0.2 for an average of 4,1 species returned, the task overall seems unreasonable.

What is the label noise of the dataset? How are Pl@ntNet image labels validated? What is the least amount of labels per species in the train set?


**Additional Feedback:**

See above.

**Clarity:**

Yes, though notation breakdown could be more clear.


**Documentation:**

Maintenance plan to be discussed.


**Ethics:**

No.

**Relation To Prior Work:**

Yes.

**Summary And Contributions:**

In this paper, the authors introduce a classification dataset built from the database of the Pl@ntNet platform. The dataset has strong class imbalance and are visually similar, which pose as challenges for set-valued classifiers. They recommend mean top-k accuracy and mean average-k accuracy as evaluation metrics and provide baseline results from deep neural networks.

---

> ### Author Response · Authors · 2021-09-28
> **Response to Reviewer iTKL (Part 1)**
>
> Thank you for your review. We are responding to your concerns as follows:
>
> > I am mainly concerned about the significance of contribution, as the main contribution of the dataset is to randomly sample 10% of genera from the Pl@ntNet database. The evaluation metrics recommended are canonical classification metrics.
>
> We believe there is a misunderstanding, the main contribution of the paper IS the Pl@ntNet database itself. Two of the authors of the paper are indeed the principal investigators of the Pl@ntNet platform and the main goal of this work was to make Pl@ntNet data available for the machine learning community.
>
> We did not publish the entire raw Pl@ntNet database (which contains unvalidated images) for two reasons:
>
> 1. It would have been too big (>500G) for widespread use by the machine learning community,
> 2. It would contain too much noise if no proper process to obtain quality data was performed.
>
> We therefore decided to publish only a portion of a high-quality subset of the entire raw database consisting of what we call the “valid” Pl@ntNet images.
>
> The label validation process is mainly based on a weighted majority voting algorithm taking as input the labels proposed by Pl@ntNet users with an adaptive weighting principle according to the user's expertise and commitment.
>
> In the new version, Section 3.1 explains the label validation processus and the technical details of this algorithm can be found in the supplementary material.
>
> The data released with this dataset is therefore new and invaluable. To give an idea of the work involved in creating, maintaining and improving an application like Pl@ntNet, let us state the following: Pl@ntNet is a 10-year project funded by public organizations. It is supported by a team of full-time engineers and researchers. It is an application used by more than 10 millions users in over 170 countries. The release of Pl@ntNet-300K is only possible because of the work done by all the people involved in this project over the years. We hope that this gives a better idea of the amount of work and time needed to develop a dataset of this magnitude.
>
> > The evaluation metrics recommended are canonical classification metrics.
>
> While top-k accuracy is a popular metric, average-k accuracy is rarely, if ever, reported in benchmarks.
>
> > In addition, there doesn't seem to be data cleaning involved.
>
> In addition to the label validation procedure described above (based on weighted majority voting), Pl@ntNet’s pipeline includes other data cleaning procedures:
> 1. automated filtering (CNN based) of unappropriated or irrelevant content (faces, humans, animals, buildings, etc.),
> 2. reporting of inappropriate content by users,
> 3. filtering on image quality (rated by users) .
>
> This was mentioned in the supplementary material of the submitted version, cf line 148-151: “The data was collected through Pl@ntNet mobile application and curated through crowdsourcing (by Pl@ntNet users) in addition to the automated filtering (CNN based) of unappropriated or irrelevant content (faces, humans, animals, buildings, etc.)”.
>
> We add this information in the new version of the paper (Section 3.1).
>
> > Though the authors retain uncertainty in the dataset, the variability involved in users taking different angled pictures of plants, different zoom/crop of plants, plants taking on different appearances depending on seasons (flowering time), is too high. Additional subsets of the dataset where this is cleaned, separated into different versions and accounted for (for example, Pl@ntNet-cropped, etc), is recommended.
>
> The Pl@ntNet-300K dataset gathers images taken by users for the purpose of plant recognition. A plant recognition system should be able to perform well regardless of the angle of the picture or the season in which the photo was taken (should we be able to recognize plants only during the flowering period, or vice versa ?). Our goal is not to publish an oversimplified dataset that does not correspond to any real world problem. We believe there are enough simple reference datasets that allow researchers to iterate quickly to evaluate their ideas (CIFAR-10, …)

---

> ### Author Response · Authors · 2021-09-28
> **Response to Reviewer iTKL (Part 2)**
>
> > The authors also acknowledge that if a person photographs only the flower of a specimen of one of the two similar species, then it will be impossible, even for an expert, to know which species the flower belongs to. Though this is a reasonable task for set-classifiers, it seems misleading then for the authors to report mean top-1 accuracy for an impossible task in Section 4.3
>
> We agree that top-1 accuracy is not the most appropriate metric in such a context. Indeed, this is the central idea of this paper and the reason why we suggest using top-k and average-k instead (with k>1). However, we believe that it is still useful to provide top-1 accuracy to highlight the limitation of this metric and motivate the need for set-valued classifiers. What's more, Figure 8-a comparing the top-1 accuracy of ImageNet and Pl@ntNet was suggested by the area chair of the previous review round.
>
> > When even the error rate for a botanist is above 0.2 for an average of 4,1 species returned, the task overall seems unreasonable.
>
> The species selected for that experiment belonged to two particularly difficult genera (with a high level of confusion).
>
> Furthermore, the performance of the botanist reflects the intrinsic difficulty of identifying plants from images alone and at the species level. When a botanist identifies a plant in the field, he or she has access to much more information than a single image (context, textures, smells) and his performance can be much better.
>
> However, the usefulness (and “reasonableness”) of image-based plant identification is no longer in question, even if it is imperfect. The number of botanists is very low and largely insufficient to cover the enormous needs in terms of identification. This taxonomic impediment has been recognized since the Rio Conference of 1992 as one of the major obstacles to the global implementation of the Convention on Biological Diversity. It is a crucial problem for the development of new approaches in many scientific fields including: biodiversity conservation and restoration, agro-ecology, land use planning, fundamental ecology, innovative governance, etc.
>
> Providing accurate and interpretable AI-based identifications is therefore of great importance for all these fields. This is why we place so much emphasis on uncertainty issues and set-valued approaches in our paper. It is precisely because the identification of a single species is too complicated a problem that we propose to approach the problem via the prediction of all possible species. Identifying a set of 5 possible species with high confidence is already useful for many scenarios, in particular because they are likely to share highly similar morphological traits. From that list, a lot of information about the ecosystem can be inferred and meaningful biodiversity monitoring indicators can be derived. Furthermore, the set of identified species could be further refined in an application like Pl@ntNet, e.g. through attribute-based approaches or geographical filtering.
>
> > What is the label noise of the dataset?
>
> Measuring label noise is a very difficult problem. Indeed, we have no oracle providing the true species of a Pl@ntNet image. We only have millions of people with heterogeneous skills, from amateurs to highly trusted experts. And even the highly trusted experts are not experts of the whole world flora. They are usually specialized on a particular flora from a particular region of the world or of a particular taxonomic group.
>
> Furthermore, label noise is highly dependent on the number of possible confusions for a given species. When two species are highly similar, they are likely to be confused more frequently.
>
> A way to measure label noise would be to ask an expert botanist to estimate it from the gallery of images of a species. However, as mentioned earlier, even a highly expert botanist will not be 100% accurate because they do not have access to missing information such as the habitat context, invisible parts of the plant, etc.
>
> > How are Pl@ntNet image labels validated?
>
> This question is reminiscent of the ApsV reviewer's comment.
> In the new version of the paper, we have added section 3.1 which deals with the label validation mechanism. The technical details can be found in the new version of the supplemental material.

---

> ### Author Response · Authors · 2021-09-28
> **Response to Reviewer iTKL (Part 3)**
>
> > What is the least amount of labels per species in the train set?
>
> If the reviewer means literally what is written, then we do not understand that question and kindly ask to rephrase it. We remind the reader that the classes are the species and that each image gets a single label (i.e. a single species).
>
> If the reviewer instead means “What is the least amount of images per label (ie per species) in the training set ?”, then the answer can be inferred from the paper. The split was described in lines 99-103 of the submitted version: “We then retain only species with more than 4 images, resulting in a total of 303 genera and L = 1,081 species. The images are divided into a training set, a validation set and a test set. For each species, 80% of the images are placed in the training set (ntrain = 243,916), 10% in the validation set (n_val = 31,118), and 10% in the test set (n_test = 31,112), with at least one image of each species in each set”. For a species with a minimum of 4 images in total, and at least one image in the validation and one image in the test set, this results in at least 2 images in the training set.
>
> We have reworked that part in the new version for a better understanding.
>
> > Maintenance plan to be discussed.
>
> The maintenance plan was detailed in the supplementary material of the submitted version, line 250-279.

---

> > ### Comment · Reviewer_iTKL · 2021-09-29
> > **Response**
> >
> > I thank the reviewers for addressing some of my concerns. Though the clarifications were helpful, I am still doubtful of the usefulness of this dataset to the wider machine learning community - re: subsets of the dataset available with more splits, re: high error rate of botanist, points other reviewers brought up, etc. I am hence slightly raising my score and acknowledge that this could be a useful dataset, but similar to ApC2, I would need to be more convinced of this dataset and benchmark's usability before publication.

---

### Official Review · Reviewer_ApC2 · 2021-09-17
**Potentially useful resource**

**Rating:** 6
**Confidence:** 5
**Clarity:** The paper is clear (apart from the co…

**Strengths:**

* Authors release a relatively novel  dataset for the classification of plants, where categories are associated to a taxonomy and where the associated classification problem is challenging.



**Weaknesses:**

 * Authors motivate the dataset as one that will allow studying set-valued classification, I think this motivation is weak (see detailed comments below), other aspects of the benchmark could be highlighted like: the inherent hierarchical classification task, the fine grained problem (mentioned only briefly), etc.

* Authors argue on the importance of set-valued metrics, but they do not provide any result of a baseline optimizing such metrics.

* I liked this dataset, but I could not determine/assess the relevance of the gap it pretends to fill, to me it looks like just another dataset (associated to an interesting problem and with their own challenges, which are not well justified) that can be introduced in a publication that describes a solution to the associated problem

**Additional Feedback:**

* Please state the differences of set-valued classification with related formulations, eg., multilabel classification, set-based classification, hierarchical, link based classification, etc.

* can you please define the right term of Eq (2)?

* Can the authors further elaborate on why it is important to approach the problem formulated in Section 2?, this is, motivation for this problem, why it is worth approaching it this way and not using multilabel classification for example?

* The information on metrics (4.1) is to detailed even when the metrics are quite common and intuitive (eq 5 is widely used as the authors mention and eq 6 is also widely used in multi label classification)

* Authors try several baselines but none of them tries to optimize the proposed evaluation metrics directly. This is one of the motivations of the dataset, yet authors still use cross-entropy as loss function. It would be interesting to know the performance of models that either optimize such metrics directly or other approaches that are appropriate for approaching this problem, e.g., methods for hierarchical classification, multilabel classification etc.


**Correctness:**

I think the claims are correct, perhaps my only critic here is that some simple concepts are presented in a complicated way and they do not help in justifying/motivating the dataset.

**Documentation:**

Yes

**Relation To Prior Work:**

This could improve, in the current version authors focus only on fine grained classification tasks. But related problems (not necessarily datasets) should be contrasted, for instance, multi label classification, set based classification, hierarchical classification etc.

**Summary And Contributions:**

Authors describe a dataset formed by 300K images of plants that are organized according to a taxonomy. The task associated to the dataset is image classification, with the main challenge that there is a high imbalance among many of the plant categories. Baselines are evaluated and compared to the performance of botanists.

---

> ### Author Response · Authors · 2021-09-27
> **Response to Reviewer ApC2 (Part 1)**
>
> Thank you for your careful review of our paper. We have updated the paper to take into account your comments. We are responding to your concerns as follows:
>
> > Authors motivate the dataset as one that will allow studying set-valued classification, I think this motivation is weak (see detailed comments below), other aspects of the benchmark could be highlighted like: the inherent hierarchical classification task, the fine grained problem (mentioned only briefly), etc.
>
> This remark echoes that of Reviewer dEbV and we agree. As the reviewer rightfully points out, the focus on set-valued classification could hide other potential uses for Pl@ntNet-300K, such as imbalanced classification under a long-tailed distribution. For this reason, we have changed the title to “Pl@ntNet-300K: a plant image dataset with high label ambiguity and a long-tailed distribution”. We have also briefly discussed in the revised manuscript the potential value of hierarchical classification for Pl@ntNet-300K.
>
> Nevertheless, we believe there is great value in applying set-valued classifiers to Pl@ntNet-300K. While for many classical datasets it is easy to obtain a high level of top-1 accuracy (MNIST, cifar10,...), making set-valued classifiers useless for these tasks, real world ambiguity makes it necessary to produce a set of candidate classes for real world tasks: Figure 5 shows that Pl@ntNet-300K mean top-1 accuracy does not exceed 40%. This reflects the difficulty of distinguishing visually similar species, see Figure 4. We argue that Pl@ntNet-300K is an ideal candidate for evaluating the performance of set-valued classifications methods: the standard method (training a modern deep neural network with cross entropy loss) yields a mean top-10 accuracy below 80% (see Figure 8). This suggests that Pl@ntNet-300K can serve as a benchmark for new set-valued classification methods. Figure 7 shows that there is a gap to close between neural networks performance and expert performance on Pl@ntNet-300K: we believe that new set-valued strategies could bridge that gap and this is what we hope to encourage by releasing Pl@ntNet-300K.
>
> We discuss these possible uses of Pl@ntNet-300K in Section 6 of the new version.
>
> > Authors argue on the importance of set-valued metrics, but they do not provide any result of a baseline optimizing such metrics.
>
> This is a very good point. Evaluating these kinds of methods is precisely one of the reasons why we release the Pl@ntNet-300K dataset. However, so far, there is no strong baseline optimizing such metrics. To the best of our knowledge, there does not exist losses designed to specifically optimize average-k accuracy. As for top-k, the only loss that we found for deep learning is the one of  Berrada et al. (2018). We have added in the supplementary material results obtained with this optimized top-k accuracy. However, the results obtained are very close to the ones obtained with cross-entropy and we believe the problem is still open to understand why and come up with new alternative methods. Our hope in releasing Pl@ntNet-300K is precisely to encourage novel methods for optimizing such metrics.
> We discuss this in Section 4.5 of the new version
>
>
> > I liked this dataset, but I could not determine/assess the relevance of the gap it pretends to fill, to me it looks like just another dataset (associated to an interesting problem and with their own challenges, which are not well justified) that can be introduced in a publication that describes a solution to the associated problem
>
> - The taxonomic impediment (i.e. the difficulty to identify living organisms) is a crucial problem for the development of new approaches in many scientific fields including: biodiversity conservation and restoration, agro-ecology, land use planning, fundamental ecology, innovative governance, etc. Providing accurate and interpretable AI-based identifications is thus of great importance for all that domains. This is why we put so much emphasis on uncertainty issues in our paper
> - Pl@ntNet dataset is an invaluable data provided by a community of more than 10M users and 2M annotators. Through this paper, we want to make this data available to the machine learning community so that they can work on data with high societal impact rather than CIFAR or ImageNet.
> - More technically speaking, Pl@ntNet dataset has the particularity to have at the same time a high amount of aleatoric and epistemic uncertainty, which makes it a benchmark candidate for a variety of tasks: long tail methods, FGVC, set-valued classification, … For a comparison of Pl@ntNet-300K to other datasets we refer the reader to Table 2.
>
> ---
>
> We respond to the other comments from Reviewer ApC2 in the next message

---

> > ### Comment · Reviewer_ApC2 · 2021-09-29
> > **R2**
> >
> > I am very grateful with the authors for clarifying some of my misconceptions on the evaluation settings. My concerns were partially addressed by reviewers. However, even with the results added by the authors, I think that there is no evidence that directly optimizing the considered metrics would yield better results than using standard classification loss functions (that is why I was asking a baseline on this). Therefore, one of the main motivations for this dataset/setting is rather weak. Still, I have raised slightly my score, I recognize the potential of the dataset but to me this benchmark is not ready for publication.

---

> > > ### Author Response · Authors · 2021-09-29
> > > **Response to Reviewer ApC2**
> > >
> > > We thank Reviewer ApC2 for his feedback on the answers we provided.
> > >
> > > Let us give some answers to the following concern:
> > >
> > > > I think that there is no evidence that directly optimizing the considered metrics would yield better results than using standard classification loss functions
> > >
> > > For experiments for which optimizing the top-k error gives better results than using the standard cross entropy, we refer the reader to the paper by Berrada et al. [1]. In particular, Table 1 of their work shows that in the presence of ambiguity, using their top-k loss (with k=5) improves top-5 accuracy .
> > >
> > > Since Pl@ntNet-300K contains a high amount of ambiguity, it is surprising that this top-k loss does not show an improvement over cross-entropy.
> > >
> > > However, let us point out that, although the problem may seem old, the topic of top-k classification is not closed:
> > >
> > > - The first top-k loss for deep learning is the one mentioned above from Berrada et al. (2018), and there is no theoretical guarantee that this loss is the best possible one, which leaves room for other methods.
> > > - Yang et al.[2] have only recently provided theoretical results on calibration and consistency (in the infinite sample limit) (2020)
> > >
> > > Regarding average-k classification, the subject is even more recent and so far the work on this topic has been purely theoretical: Denis [3]  (2017), Lorieul [4] (2020).
> > >
> > > Thus we believe set-valued classification is still an open subject and argue that P@ntNet-300K is a dataset for which these issues are of the utmost importance, making it a relevant candidate for evaluation and validation of future methods.
> > >
> > > That being said, we hope that the changes made to the paper and discussions with the reviewers during this phase have convinced readers that the value of Pl@ntNet-300K goes beyond set-valued classification.
> > >
> > > We would like to thank Reviewer ApC2 again for his relevant comments.
> > >
> > > ---
> > >
> > >
> > > [1] Smooth Loss Functions for Deep Top-k Classification - Berrada et al. (2018)
> > >
> > > [2] On the consistency of top-k surrogate losses - Yang, Koyejo (2020)
> > >
> > > [3] Confidence Sets with Expected Sizes for Multiclass Classification - Denis, Hebiri (2017)
> > >
> > > [4] Uncertainty in predictions of deep learning models for fine-grained classification - Lorieul (2020)

---

> ### Author Response · Authors · 2021-09-27
> **Response to Reviewer ApC2 (Part 2)**
>
> > I think the claims are correct, perhaps my only critic here is that some simple concepts are presented in a complicated way and they do not help in justifying/motivating the dataset.
>
> This remark is reminiscent of the comment made by Reviewer dEbV. While top-k classification is popular, average-k classification is not widespread, and therefore we believe it is necessary to introduce the framework in Section 2.
>
> > Relation To Prior Work: This could improve, in the current version authors focus only on fine grained classification tasks. But related problems (not necessarily datasets) should be contrasted, for instance, multi label classification, set based classification, hierarchical classification etc.
>
> We thank Reviewer ApC2 for allowing us to clear up a confusion that may be shared by some readers.
>
> In our setting, introduced in Section 2, there is a single class associated with an image. Indeed, users are asked to photograph a specimen of a plant and thus the vast majority of images consist of a single specimen of a plant. This is transcribed on line 56 of the submitted version : $Y  \in \{1, …, L\}$.
>
> This differs from the multi label setting, where multiple labels can be associated with an image (for instance MS COCO, Pascal VOC, …). We insist that a single label is provided for each image.
>
> We believe that it is important that the reader understands the difference between the two settings and thank Reviewer ApC2 for allowing us to clarify that point in the new version. This is detailed in Section 2 of the new version. Regarding hierarchical classification, we wrote earlier in this answer that we do not make use of the genera. This is specified in the new version in Section 6.
>
> > can you please define the right term of Eq (2)?
>
> The right term of equation 2 was defined on line 70 of the submitted version.
>
> > Can the authors further elaborate on why it is important to approach the problem formulated in Section 2?, this is, motivation for this problem, why it is worth approaching it this way and not using multilabel classification for example?
>
> As explained above, we are not in the multi-label setting as there is only one true label per image.
>
> To answer the reviewer's question, consider the case of the Pl@ntNet application, where a user takes a picture of a plant specimen and wants to know what species it is.
>
> For some images the model will have no problem giving the correct species (consider a common species visually very different from other species).
>
> However, for some images it might be complicated (if not impossible) for a model to give the true class with certainty. Consider for instance Figure 4:  for each image, it is reasonable to suggest to the user at least two species, if not more.
>
> Top-k classification and average-k classification allow to return a set of species, only they differ in the way they do so: in the top-k case, k species are systematically returned. In the average-k setting, the number of classes returned is not fixed and depends on the image. However, k classes are returned on average, hence the name.
>
> We transcribe this discussion in Section 2 of the new version.
>
> > The information on metrics (4.1) is to detailed even when the metrics are quite common and intuitive (eq 5 is widely used as the authors mention and eq 6 is also widely used in multi label classification)
>
> We drop the top-k accuracy definition in the new version. However, as explained above, we are not in the mult-label setting and we believe that average-k accuracy is much less common than top-k accuracy in the multi-class classification setting, so we keep its definition.
>
> > Authors try several baselines but none of them tries to optimize the proposed evaluation metrics directly. This is one of the motivations of the dataset, yet authors still use cross-entropy as loss function. It would be interesting to know the performance of models that either optimize such metrics directly or other approaches that are appropriate for approaching this problem, e.g., methods for hierarchical classification, multilabel classification etc.
>
> This was answered in the first part of our message
>
> ---
>
> We thank the Reviewer ApC2 again for his helpful comments

---

### Official Review · Reviewer_ApsV · 2021-09-19
**A good dataset for evaluating set-valued classifiers and long-tailed learning but lacking some minor details**

**Rating:** 6
**Confidence:** 4

**Strengths:**

-	The paper proposes a big enough dataset for experimentation with ML algorithms. The dataset could benefit the research community.
-	The authors analyzed the relationship of top-k accuracy and mean-k accuracy. It showed that their relationship is not trivial.
-	Relationships between the performance on the ImageNet dataset and the proposed dataset of baseline models are shown.
-	The discussions of Epistemic and Aleatoric uncertainties are well written.


**Weaknesses:**

-	The dataset contains species that have at least four images. It is split into train, valid, and test with a ratio of 8:1:1. It is unclear how the author deals with species with a small number of images (4-5 images). How many are in the train, valid and test set for such cases?
-	Section 4.2 Baseline lacks training details like training epoch and learning rate schedule.
-	The performances of baselines on average-k metrics were not shown in the paper. Figure 6 has mean average-5 accuracy as the y axis, but the caption is top-5 accuracy.
-	The reviewer wonders how many annotators participated in labeling the original dataset and their consensus. The paper would benefit from the discussion regarding inter-annotator consensus in section 3.3 Aleatoric uncertainty.


**Additional Feedback:**

According to the authors, which metrics are the best to measure the performance of a model on this dataset, top-k accuracy or average-k accuracy?

**Clarity:**

Although some minor details are missing, the paper is well written and easy to follow.

**Correctness:**

-	The proposed dataset is created from a bigger dataset such that the nature of the original dataset is kept, i.e., long-tailed distribution, intra-class variability, and ambiguous data points.
-	The proposed metrics: Top-k accuracy and Average-k accuracy, are suitable for evaluating set-valued classifiers.


**Documentation:**

The dataset is well documented: from the creation process to the process of acquiring it.
-	There is a link in the paper that leads to a download site for the dataset.
-	The author also published experiments code on GitHub.


**Ethics:**

The proposed dataset is created from a much bigger public plant dataset, so there is no ethical concern.

**Relation To Prior Work:**

The author compares the proposed dataset with other related datasets on Fined-Grained Visual Categorization in many attributes such as long-tailed, intra-class variability, and others.

**Summary And Contributions:**

The paper presents a long-tailed image dataset with high intra-class variability which is big enough (306,146 images) for experimentation with ML techniques, and contains many types of uncertainty. The authors further proposed performance metrics to evaluate models trained on the dataset that accounts for its uncertainties. The author also provides many baselines and analyses into the relationship between proposed metrics.

---

> ### Author Response · Authors · 2021-09-27
> **Response to Reviewer ApsV**
>
> Thank you for carefully reviewing our document and pointing out typos and information we forgot to add. We have updated the paper to take into account your comments. We address your concerns as follows:
>
> > The dataset contains species that have at least four images. It is split into train, valid, and test with a ratio of 8:1:1. It is unclear how the author deals with species with a small number of images (4-5 images). How many are in the train, valid and test set for such cases?
>
> When there are a small number of images per class (4-5), we place 1 image in the validation set, 1 image in the test set, and the rest in the training set. This is what we meant by “with at least one image of each species in each set”. More formally, n_test = ceil(0.1 x n_class), n_val = ceil(0.1 x n_class), n_train = n_class - n_val - n_test.
>
> > Section 4.2 Baseline lacks training details like training epoch and learning rate schedule.
>
> Thank you for reporting this oversight on our part. In the new version, we add all the hyperparameters used to produce the results.
>
> > The performances of baselines on average-k metrics were not shown in the paper. Figure 6 has mean average-5 accuracy as the y axis, but the caption is top-5 accuracy.
>
> Thank you for pointing this out. The caption should indeed be “mean average-5 accuracy”. We have corrected the typo in the updated version.
>
> > The reviewer wonders how many annotators participated in labeling the original dataset and their consensus. The paper would benefit from the discussion regarding inter-annotator consensus in section 3.3 Aleatoric uncertainty.
>
> Label validation is based on a weighted majority voting algorithm taking as input the labels proposed by Pl@ntNet users with an adaptive weighting principle according to the user's expertise and commitment.
> Thus, a single trusted annotator can be enough to validate an image label.
> On the other hand, images whose labels are proposed by several novice users may not be validated because they do not have sufficient weight.
>
> The technical details of this algorithm can be found in the supplementary material (updated version).
>
> At the time of the construction of Pl@ntNet-300K, the total number of annotators in the Pl@ntNet database was equal to 2,079,003.
> The average number of annotators per image is equal to 2.03 but this number can vary significantly.
>
> We added this discussion in the paper in Section 3.1 and provided additional technical details in the supplementary materials.
>
> > According to the authors, which metrics are the best to measure the performance of a model on this dataset, top-k accuracy or average-k accuracy?
>
> While average-k accuracy and top-k accuracy metrics are correlated (see Figure 8), they reflect different scenarios.
>
> Top-k accuracy models the case where a system always returns k images to a user (say k=3). This metric is popular and is traditionally reported when evaluating methods on ImageNet. However, this setting is somewhat unsatisfactory: consider the case where a user takes a picture of a very common and distinctive species. The model will easily recognize the true label so it might not be necessary to return all k species to the user when the model is so confident: a single suggestion would suffice.
>
> However, for some species the task of classifying a specimen is much more difficult, (see Figure 4) and in these cases, returning only k candidate classes might not be sufficient: we would like the user to know all the likely species for that photograph. This is what average-k classification allows us to do: provide a set of candidate classes of different sizes.
>
> Following your question, we have expanded Section 4.4.

---

### Official Review · Reviewer_dEbV · 2021-09-21
**Big Image classification dataset to identify species of a plant.**

**Rating:** 7
**Confidence:** 3
**Correctness:** seems correct.
**Clarity:** Yes, the paper is well written.

**Strengths:**

- A huge dataset on a domain which is not covered much in other image datasets.
- A good set of experiments with recent classification models were done to showcase the difficulties in the dataset.
- Experiments shows that state-of-art techniques has a lot more gaps to fill to reach human annotator accuracy.
- Authors seems to have addressed all the concerns raised by the meta-reviewer in the first round (this is what I learned from the supplementary material.)


**Weaknesses:**

- I think the "Plant Village dataset"[1] should be cited in this paper. It is a plant image classification dataset with plant diseases and species as classes. If there is an overlap of species between these datasets, can they be combined to create better classifiers?
- Writing:
    - Set-valued classifiers seems to be a recent paper and it is not very clear why this particular dataset be the best case for that experimental setup. Having this experimental protocol in the title is taking the focus away from the dataset.
    - There are much more details on algorithm and loss functions provided, which personally I feel does not fit well in a dataset paper.


[1] Sladojevic, Srdjan, et al. "Deep neural networks based recognition of plant diseases by leaf image classification." Computational intelligence and neuroscience 2016 (2016).

**Additional Feedback:**

NA

**Documentation:**

The authors provide adequate details on the dataset and the evaluation setup.

**Ethics:**

No ethical concerns.

**Relation To Prior Work:**

Missed an important reference on "Plant Village" dataset missing. Refer to "Weaknesses".

**Summary And Contributions:**

The paper introduces a challenging image classification dataset with 300K+ plant images grouped into 1081 classes (species). This dataset has difficulties which is often observed in practical applications of Image classification. The dataset is so difficult that an expert human annotator gets only 80% times correct. The species are grouped in to Genus, so there is a hierarchical structure in the way classes are related. Number of instances per classes is skewed which brings a huge class-imbalance issue which makes the dataset more challenging.

---

> ### Author Response · Authors · 2021-09-27
> **Response to Reviewer dEbV**
>
> Thank you for your thorough review of our paper. We have updated the paper to take into account your comments. We are responding to your concerns as follows:
>
>
> > I think the "Plant Village dataset"[1] should be cited in this paper. It is a plant image classification dataset with plant diseases and species as classes. If there is an overlap of species between these datasets, can they be combined to create better classifiers?
>
> We thank you for mentioning the “Plant Village dataset”. Table 1 of this paper indicates that there are only 15 classes (including the background class) against 1081 for Pl@ntNet. Besides, there are only 4483 original images versus 306,146 for Pl@ntNet-300K. More importantly, the class overlap appears to be null.  Hence, combining the two datasets would not bring much power to learning algorithms.
>
> > Set-valued classifiers seems to be a recent paper and it is not very clear why this particular dataset be the best case for that experimental setup. Having this experimental protocol in the title is taking the focus away from the dataset.
>
> We agree with the reviewer that the title and emphasis on set-valued classifiers could distract from the point of the dataset: a new dataset with high aleatoric and epistemic uncertainty. This remark goes hand in hand with that of Reviewer ApC2. This is why we have changed the title to “Pl@ntNet-300K: a plant image dataset with high label ambiguity and a long-tailed distribution”.
>
> However, we believe that there is great value in applying set-valued classifiers to such ambiguous datasets. While for many classical datasets it is easy to obtain high level of top-1 accuracy (MNIST, CIFAR-10,...), rendering set-valued classifiers useless for these tasks, real world ambiguity makes it necessary to produce a set of candidate classes for real world tasks: Figure 5 shows that Pl@ntNet-300K mean top-1 accuracy is no more than 40%. This reflects the difficulty of distinguishing visually similar species, see Figure 4. We argue that Pl@ntNet-300K is an ideal candidate for evaluating the performance of set-valued classifications methods: the standard method (training a modern deep neural network with cross entropy loss) yields a mean top-10 accuracy below 80% (see Figure 8). This suggests that Pl@ntNet-300K can serve as a benchmark for new set-valued classification methods. Figure 7 shows that there is a gap to close between neural networks performance and expert performance on Pl@ntNet-300K: we believe that new set-valued strategies could bridge that gap and this is what we hope to encourage by releasing Pl@ntNet-300K.
>
> As the reviewer rightfully pointed out, the focus on set-valued classification may obscure other potential uses for Pl@ntNet-300K, such as imbalanced classification under a long-tailed distribution. We have modified the title accordingly and added a discussion in the paper in Section 6. As mentioned by reviewer ApC2, hierarchical classification could also be an interesting challenge and we mentioned it as well.
>
> > There are much more details on algorithm and loss functions provided, which personally I feel does not fit well in a dataset paper.
>
> Thank you for sharing your concern on notation and loss functions. This is something that we are aware of. However, while top-k classification is widespread, average-k is, as you mentioned in your previous remark, relatively new and thus less popular. With this paper, we hope to draw attention to this problem, and thus we think it is necessary to introduce the framework in Section 2 and the corresponding metrics in Section 4

---

### Author Response · Authors · 2021-09-29
**Summary of paper modifications**

We thank the reviewers for their constructive feedback and for providing helpful comments to improve the paper.
We list here the changes we made to the paper in response:

1. In the new version, we clearly mention potential uses of Pl@ntNet-300K other than evaluating set-valued classification methods in Section 6, such as imbalanced classification under a long-tailed distribution, Fined Grained Visual Categorization, or even hierarchical classification. We also change the title accordingly: “Pl@ntNet-300K: a plant image dataset with high label ambiguity and a long-tailed distribution”.
2. A section on label validation and data cleaning is added in Section 3.1
3. The motivation for the need for set-valued classifiers is further detailed in Section 2 of the new version. In this same section, we emphasize that our framework is different from multi-label classification.
4. We add Section 4.5 which discusses the evaluation of existing set-valued classification techniques on Pl@ntNet-300K. We emphasize that our hope in publishing Pl@ntNet-300K is precisely to encourage new methods for optimizing such metrics
5. The distribution between the training set, the validation set and the test set is more detailed in the new version
6. A table containing model-specific hyperparameters is added in Section 4.2 (Table 1)
7. The reference to the Plant Village dataset is added in the new version
8. We drop the definition of top-k accuracy in Section 4
9. The caption of Figure 6 is fixed

**More generally, we would like to emphasize that the Pl@ntNet dataset is an invaluable data provided by a community of more than 10 million  users and 2 million annotators. Through this paper, we want to make this data available to the machine learning community so that they can work on data with high societal and environmental impact.**

We refer the reader to the responses provided separately to the reviewers for an in-depth treatment of their questions.

---

### Decision · Program_Chairs · 2021-10-09

**Decision:**

Accept

**Comment:**

Although the dataset shows unique aspects, there are critiques related to lack of proper motivation, missing design decisions (noise, sampling decisions, partitions, labeling, sanity check, etc.) and analyses. However after discussion with the authors, several critiques have been clarified. Overall the paper achieves the minimum score to be accepted for publication at NeurIPS data track.